# Short contacts between chains enhancing luminescence quantum yields and carrier mobilities in conjugated copolymers

Tudor H. Thomas[1,4], David J. Harkin[1,4], Alexander J. Gillett[1], Vincent Lemaur[2], Mark Nikolka[1], Aditya Sadhanala[1], Johannes M. Richter[1], John Armitage[1], Hu Chen[3], Iain McCulloch[3], S. Matthew Menke[1], Yoann Olivier [2], David Beljonne[2] & Henning Sirringhaus [1]

Efficient conjugated polymer optoelectronic devices benefit from concomitantly high luminescence and high charge carrier mobility. This is difficult to achieve, as interchain interactions, which are needed to ensure efficient charge transport, tend also to reduce radiative recombination and lead to solid-state quenching effects. Many studies detail strategies for reducing these interactions to increase luminescence, or modifying chain packing motifs to improve percolation charge transport; however achieving these properties together has proved elusive. Here, we show that properly designed amorphous donor-*alt*-acceptor conjugated polymers can circumvent this problem; combining a tuneable energy gap, fast radiative recombination rates and luminescence quantum efficiencies >15% with high carrier mobilities exceeding 2.4 cm$^2$/Vs. We use photoluminescence from exciton states pinned to close-crossing points to study the interplay between mobility and luminescence. These materials show promise towards realising advanced optoelectronic devices based on conjugated polymers, including electrically-driven polymer lasers.

[1] Cavendish Laboratory, University of Cambridge, JJ Thomson Avenue, Cambridge CB3 0HE, UK. [2] Université de Mons, Place du Parc 20, 7000 Mons, Belgium. [3] KAUST Solar Center (KSC), KAUST, Thuwal 23955-6900, Saudi Arabia. [4] These authors contributed equally: Tudor H. Thomas, David J. Harkin. Correspondence and requests for materials should be addressed to H.S. (email: hs220@cam.ac.uk)

Conjugated polymer semiconductors have attracted significant attention in recent years, and materials that combine high charge carrier mobility ($\mu$) and fluorescence quantum efficiency ($\Phi$) are sought after for optoelectronic applications[1,2]. Polymer organic light-emitting diodes (OLEDs) and organic photovoltaic devices (OPVs) benefit from the tune-ability of the energy gap[3,4], high solubility[4,5], operational stability[5,6], and the uniformity that these often near amorphous polymer films provide. However currently, materials in state-of-the-art devices are usually selected for their individually opti-mised $\Phi$ or $\mu$ values[7]. Materials that have both high $\Phi$ and $\mu$ are uncommon. This has hampered efforts to simplify device archi-tectures[8], and to achieve high brightness OLEDs towards electrically-driven organic lasing applications[9]. There is an apparent trade-off between these two desirable properties.

Central to this trade-off is the degree of interchain interaction. Strong electronic coupling mediated by extended and close con-tacts between chains is thought to be crucial for realising high $\mu$ in amorphous polymer systems where charge transport is limited by the ability to hop between adjacent chains[10,11]. However, strong interchain interactions have been shown empirically to lead to low $\Phi$. This has been attributed to a number of different mechanisms, including increased exciton diffusion to pre-existing chemical defects or exciton quenching 'trap' sites[12,13], an addi-tional density of charge-transfer interchain states with low oscillator strength[13–16], additional fast non-radiative recombina-tion channels arising from intersecting bands[17–19], and the sup-pression of radiative recombination pathways by H-aggregate formation[20].

## Results

Another aspect of this trade-off is the energy gap ($E_g$) of the polymer. $\Phi$ generally increases with $E_g$, and this trend is shown for various polymers in Fig. 1a, and tabulated values are provided in Supplementary Table 1. The energy gap law predicts fast non-radiative recombination in low $E_g$ materials[21], while the rate of radiative (spontaneous) emission increases with $E_g$[22]. Therefore, low energy gap polymers (with $E_g < 1.8$ eV) typically have low $\Phi$, and the most fluorescent polymers— including poly(9,9-di-n-octylfluorenyl-2,7-diyl) (F8, $\Phi = 0.65$, $E_g = 3.0$ eV[23]) and poly(9,9-dioctylfluorene-*alt*-benzothiadiazole) (F8-BT, $\Phi = 0.5$, $E_g = 2.5$ eV[24])—have $E_g > 2.0$ eV. On the other hand, in Fig. 1a, $\mu$ tends to decrease with increasing $E_g$: While F8 and F8-BT have low $\mu < 10^{-2}$ cm²/Vs, polymers with $\mu > 1$ cm²/Vs such as diketopyrrolopyrrole-derivatives[25,26], and poly(bis(3-alkylthio-phen-2-yl)thienothiophene) (pBTTT)[27] have $E_g < 1.8$ eV and low $\Phi$. This does not reflect a direct relationship between $\mu$ and $E_g$, but is likely to be due to limitations in charge injection, or increased charge carrier trapping due to atmospheric species in polymers with deeper ionisation potentials and larger $E_g$[28,29]. Interestingly, in small molecule semiconductor systems high $E_g$, $\mu$ and $\Phi$ have been observed simultaneously[30].

Recently, a family of high mobility donor-acceptor copolymers with weak structural ordering, a near amorphous microstructure, but a low degree of energetic disorder, and high charge carrier mobility has attracted significant attention[27]. Polymers such as poly(indacenodithiophene-*alt*-benzothiadiazole) (IDTBT in the literature, but hereunder IDT-H$_2$BT) exhibit high $\mu \sim 2$ cm²/Vs; but with $\Phi < 0.02$ ($E_g = 1.8$ eV). Such a low fluorescence quantum efficiency prevents the use of this family of high mobility polymers in optoelectronic applications. In this study, we investigate whe-ther the low quantum yield is an inherent feature, or whether it can be improved by molecular design without sacrificing mobility.

The success of IDT-H$_2$BT as an amorphous high-mobility polymer semiconductor can be attributed to a high potential

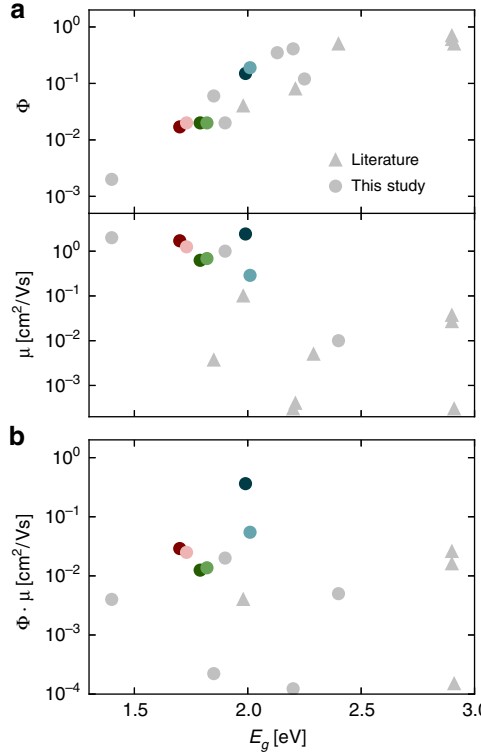

**Fig. 1** Relationship between mobility and luminescence in conjugated polymers. **a** Quantum yield $\Phi$ and charge carrier mobility $\mu$ (extracted from FET measurements) against bandgap energy $E_g$; **b** Product $\Phi \cdot \mu$ parametrised in terms of the $E_g$ of different materials: literature (triangles) and this study (circles)

barrier to torsion between donor and acceptor sub-units. This arises from a variety of reasons, the most important of which seems to be the strong non-covalent interaction between the hydrogen (on the alpha carbon) of the IDT with the nitrogen from BT; resulting in a stiff backbone with a narrow density of states and a low degree of energetic disorder[27]. Insertion of a thiophene or benzene ring elongating the donor subunit retains the low disorder character and yields poly(indacendithienothio-phene-*alt*-benzothiadiazole) (IDTT-H$_2$BT)[31] or poly(dithiophene indenofluorene-*alt*-benzothiadiazole) (TIF-H$_2$BT)[32], respectively. The chemical structures of these polymers are shown in Fig. 2a. We are also investigating the perfluorinated-benzothiadiazole (F$_2$BT) analogues. To generally evaluate materials for optoelec-tronic applications mentioned above, we assign equal importance to $\Phi$ and $\mu$, and use as our figure of merit their product. Figure 1b combines the highest published values for polymers with the materials reported in this study. To the best of our knowledge, the highest $\Phi \cdot \mu$ of $2.6 \times 10^{-2}$ cm²/Vs in the literature was reported for an F8 derivative[1].

Here, we show that TIF-H$_2$BT reaches a more than an order of magnitude higher value of $\Phi \cdot \mu = 0.36$ cm²/Vs. By inserting fused rings into the backbone and increasing the interchain interaction at close-crossing points, it thus seems possible to simultaneously increase mobility and fluorescence. We report a careful study of the underpinning photophysics combining time-resolved spec-troscopy with quantum-chemical modelling.

**Electrical characterisation**. We use organic field-effect transistor (OFET) devices to confirm that all investigated materials have a high *p*-type mobility comparable to (or exceeding) that of amorphous-Si[33]. Care was taken not to overestimate the mobility

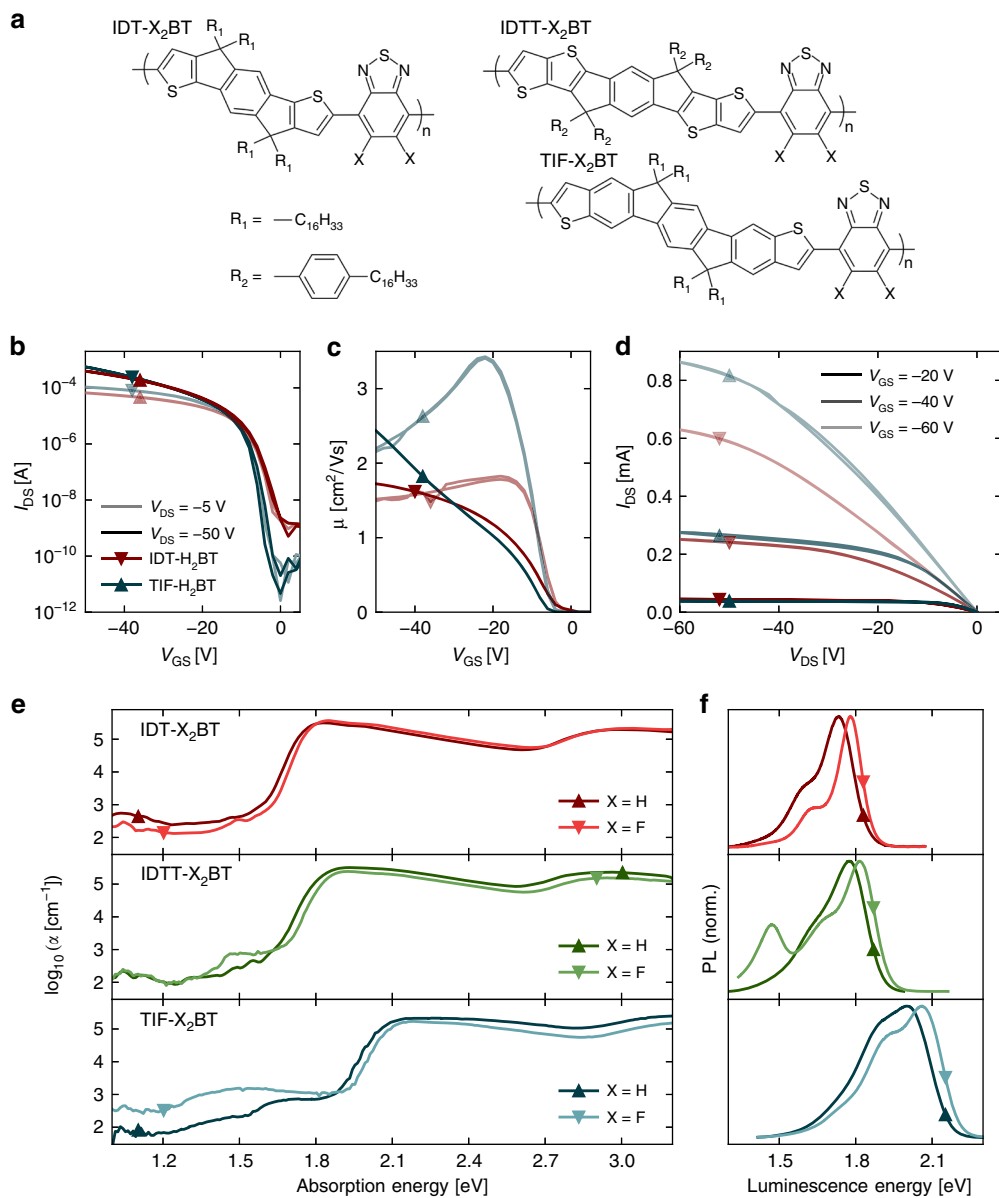

**Fig. 2** Charge transport and optical properties of polymers in this study. **a** Structures of IDT-X$_2$BT, IDTT-X$_2$BT, and TIF-X$_2$BT with X = H and F. Transfer characteristics (**b**), gate voltage dependence of mobilty (**c**) and output characteristics (**d**) for IDT-H$_2$BT (red) and TIF-H$_2$BT (blue). **e** PDS absorption and (**f**) normalised PL spectra of the six polymers

of our optimised devices where they deviate from MOSFET behaviour. Meaningful extraction of the mobility is discussed in detail in Supplementary Note 1, and by Bittle et al.[34]. In Fig. 2b, we compare the two highest-performing materials, IDT-H$_2$BT (red) and TIF-H$_2$BT (blue) in linear (light) and saturation (dark) regimes. Low threshold voltages, sharp subthreshold swings and on:off-current ratios ~$10^5$–$10^7$ indicate good charge injection into the polymers with low trap densities for both materials. Furthermore, high on-currents approaching 1 mA (with a channel width:length ratio of 50 and a gate capacitance of 3.7 nF/cm$^2$) are measured for both polymers. The on-current of TIF-H$_2$BT is higher by about 50% than that of IDT-H$_2$BT, clearly indicating a higher $\mu$.

Extraction of $\mu$ as a function of gate voltage ($V_{GS}$) is shown in Fig. 2c in the linear ($V_{DS} = -5$ V) and saturation ($V_{DS} = -50$ V) regimes within the MOSFET model. For both materials, contact resistance remains sufficiently low and the output characteristics in Fig. 2d lack a sigmoidal contact artefact near $V_{DS} = 0$ V. For

IDT-H$_2$BT, there is little deviation from ideal MOSFET behaviour, and it shows a field-effect mobility exceeding 1.7 cm$^2$/Vs, independent of gate voltage in the linear and saturation regime. TIF-H$_2$BT was less ideal in its OFET characteristics, and exhibited a significant dependency of the field-effect mobility on the gate voltage. We conservatively estimate a mobility of 2.4 cm$^2$/Vs from both the linear and saturation regimes.

By comparison, we observe a moderate decrease in $\mu$ to ~1.3 cm$^2$/Vs for IDT-F$_2$BT. IDTT-H$_2$BT and IDTT-F$_2$BT have similar $\mu$ ~ 0.63 − 0.69 cm$^2$/Vs, and the lower $\mu$ here is most likely due to an unfavourable increased chain stacking distance due to increased steric hindrance induced by hexadecylphenyl side-chains, as seen before for IDT-H$_2$BT[35]. Finally, we observe a substantial decrease in $\mu$ between TIF-H$_2$BT and TIF-F$_2$BT, and the latter has lowest $\mu = 0.29$ cm$^2$/Vs. The high ionisation potential of TIF-F$_2$BT (of 5.8 eV) leads to increased contact resistance (Supplementary Fig. 1), and also leaves holes

particularly vulnerable to deep atmospheric and water-related trap states[32,36]. The OFET data for these materials are shown in Supplementary Fig. 2 with tabulated values in Supplementary Table 2.

**Optical characterisation**. Absorption spectra of polymer films are shown in Fig. 2e. Using photothermal deflection spectroscopy[37], we measure the absorption coefficient ($\alpha$) over four orders of magnitude in intensity. The $E_g$ is defined where $\alpha$ decreases quickly by a factor of $10^2$–$10^3$, and (approximately) corresponds in energy to the on-chain (internal) charge transfer state (ICT) where the hole is localised on the donor and the electron is localised on the acceptor[38]; we clarify this definition in the modelling section. We observe that increasing the length of the donor unit from IDT-H$_2$BT to IDTT-H$_2$BT to TIF-H$_2$BT increases the $E_g$ as it decreases the relative push-pull character of the repeat unit, and that substitution of H$_2$BT (dark) for F$_2$BT (light) slightly increases $E_g$ for each polymer, a phenomenon well-documented for comparable systems[39,40]. We observe the same trends in the photoluminescence (PL) (Fig. 2f), leading to a low Stokes shift of ~100 meV. Further discussion is provided in Supplementary Note 2.

By fitting the $\alpha$ tail over the $E_g$ edge with Eq. 1;

$$\alpha(\hbar\omega) \sim \exp\left(\frac{\hbar\omega - E_g}{E_U}\right) \qquad (1)$$

we extract the disorder in the joint (J) density of states (DOS) via the Urbach energy ($E_U$)[41]. This has been shown to correlate closely with energetic disorder in the transport DOS for polymers[42], and the low $E_U$ in IDT-H$_2$BT—despite the lack of long-range structure—is crucial for MOSFET-like behaviour[27]. We find that increasing the length of the donor subunit marginally increases $E_U$ from $25 \pm 2$ meV for IDT-H$_2$BT, to $27 \pm 2$ meV for IDTT-H$_2$BT and $31 \pm 3$ meV for TIF-H$_2$BT. These values did not change upon substitution of F$_2$BT. The extracted energies here are among the lowest reported values for any polymer semiconductors; even lower than the best performing semi-crystalline materials[27]. The slightly higher value measured for TIF-H$_2$BT is potentially consistent with the gate voltage dependence of the mobility being more pronounced than in IDT-H$_2$BT.

We have measured the luminescence quantum efficiency using an integrating sphere. The differences observed in $\Phi$ in thin films are profound, and follow the energy gap law. While IDT-H$_2$BT has the lowest $\Phi$ of 0.017, elongation of the backbone and substitution of F$_2$BT increases $\Phi$ monotonically to the high values of 0.15 and 0.19 for TIF-H$_2$BT and TIF-F$_2$BT, respectively. This results in a record value $\Phi \cdot \mu = 0.36$ cm$^2$/Vs for TIF-H$_2$BT and demonstrates clearly that the extension of the conjugation length from IDT-H$_2$BT to TIF-H$_2$BT is associated with a significant increase of both mobility and luminescence quantum efficiency. We estimate the exciton diffusion length to be $12.8 \pm 1.9$ nm in TIF-H$_2$BT using a method described elsewhere[12]; an improvement over $7.1 \pm 1.1$ nm for IDT-H$_2$BT, and therefore a greater H-aggregate character, as described in Supplementary Note 2. This suggests that the more extended conjugation and relatively smaller sidechain density of this polymer facilitate more pronounced interchain interactions at close-crossing points of the polymer chains, which are fully consistent with the observed increase in field-effect mobility.

**Long-lived emission from interchain species**. In some of the polymers, we observe luminescence that is not associated with the ICT state. This is most directly apparent in IDTT-F$_2$BT where the additional PL peak below 1.5 eV cannot be explained by the

vibronic progression of the ICT. This redshifted emission pathway is not present in dilute solution measurements, but rises upon addition of non-solvent precipitating suspended aggregates in solution[43]. It is also decreased upon blending the polymer with a PMMA matrix, and we can tune the electroluminescence ratio of the two pathways with current densities in IDTT-H$_2$BT OLED architectures (Supplementary Note 3, Supplementary Figs. 3 and 4). This suggests the formation of emissive interchain or aggregate species. Such species have been observed before in neat polymer systems, and as such, they are often assumed to be relatively charge-separated excitations and assigned as interchain, charge transfer species (CTs) or pairs of polarons (PPs)[15,44–47]. However, in our case we find that their precise nature is more subtle. Therefore, we refer to such species loosely (for now) as (aggregate) interchain species (IS), since they only form in the presence of interchain contacts (i.e., they disappear in dilute solution and in dilute blends). We will clarify the nature of the IS species later in the modelling section. Notwithstanding, it is uncommon for such interchain species in polymers to be emissive. Luminescence from inter-molecular CTs in OPV systems is also uncommon[48,49]; however it has been used as a tool to probe the bulk heterojunction interfaces[50]. We use the IS emission here in a similar way, as a probe of the close-crossing points in the polymer film that critically influence the carrier mobility and luminescent properties. It is interesting to note that the additional luminescence has a corresponding sub-$E_g$ absorption feature at 1.50 eV in IDTT-F$_2$BT, which is not present for IDTT-H$_2$BT in Fig. 2e. Similar sub-$E_g$ features are observed in IDT-H$_2$BT, TIF-H$_2$BT and TIF-F$_2$BT, and discussed in Supplementary Note 3 (Supplementary Figs. 5 and 6).

A direct way to search for IS emission signatures is to measure time-resolved photoluminescence, as interchain species presumably have different (longer) recombination lifetimes than their ICT. Using time-correlated single photon counting (TCSPC; $E_{pump} = 3.05$ eV, $f_{pump} \sim 2$ $\mu$J/cm$^2$), we measure the decay kinetics of the PL over the emissive band at different detection wavelengths. For IDTT-F$_2$BT, we observe that the IS emission has a longer lifetime ($\tau_{IS}$) of 760 ps compared with the ICT ($\tau_{ICT}$), which is faster than the 250 ps instrument response. This allows for the spectral deconvolution of the two species based on their different kinetics, shown for some polymers in Fig. 3a, using a previously described genetic algorithm[51], and a methodology discussed in Supplementary Note 3.

To complete the picture of IS luminescence in this polymer family, we determine the timescales of formation of these states in Fig. 3c using ultrafast transient grating PL. For TIF-H$_2$BT, in the first 1 ps after excitation, its broad PL spectrum narrows and redshifts as the unequilibrated, 'hot' ICT population (ICT*) thermalises to the ICT. By 5 ps, two luminescence bands appear; as the low-energy PL tail (due to IS states) decays more slowly than the ICT. By integrating the shaded spectral windows, we show the normalised decay kinetics of three colour-corresponding energy bands in Fig. 3e, which, to a good approximation, denote ICT* (dark grey), ICT (cyan) and IS (sky blue) decay, respectively. To within 100 fs, all three bands rise at the same time, so we conclude that the interchain species must be formed at least as fast as the ICT* relaxation (Supplementary Note 3). Ultrafast formation of interchain species, and the following recombination proceeding on multiple timescales has also been measured in other neat polymer systems[52–58].

In general, we also expect energy transfer from the ICT to lower-lying IS on timescales >1 ps. To determine the yield of this transfer, we use ultrafast transient absorption (TA) spectroscopy. Representative normalised TA spectra for TIF-H$_2$BT ($E_{pump} = 2.34$ eV, $f_{pump} = 25$ $\mu$J/cm$^2$, $n_0 = 5 \times 10^{18}$ cm$^{-3}$) are shown in Fig. 3d, but all six materials behave in the same way

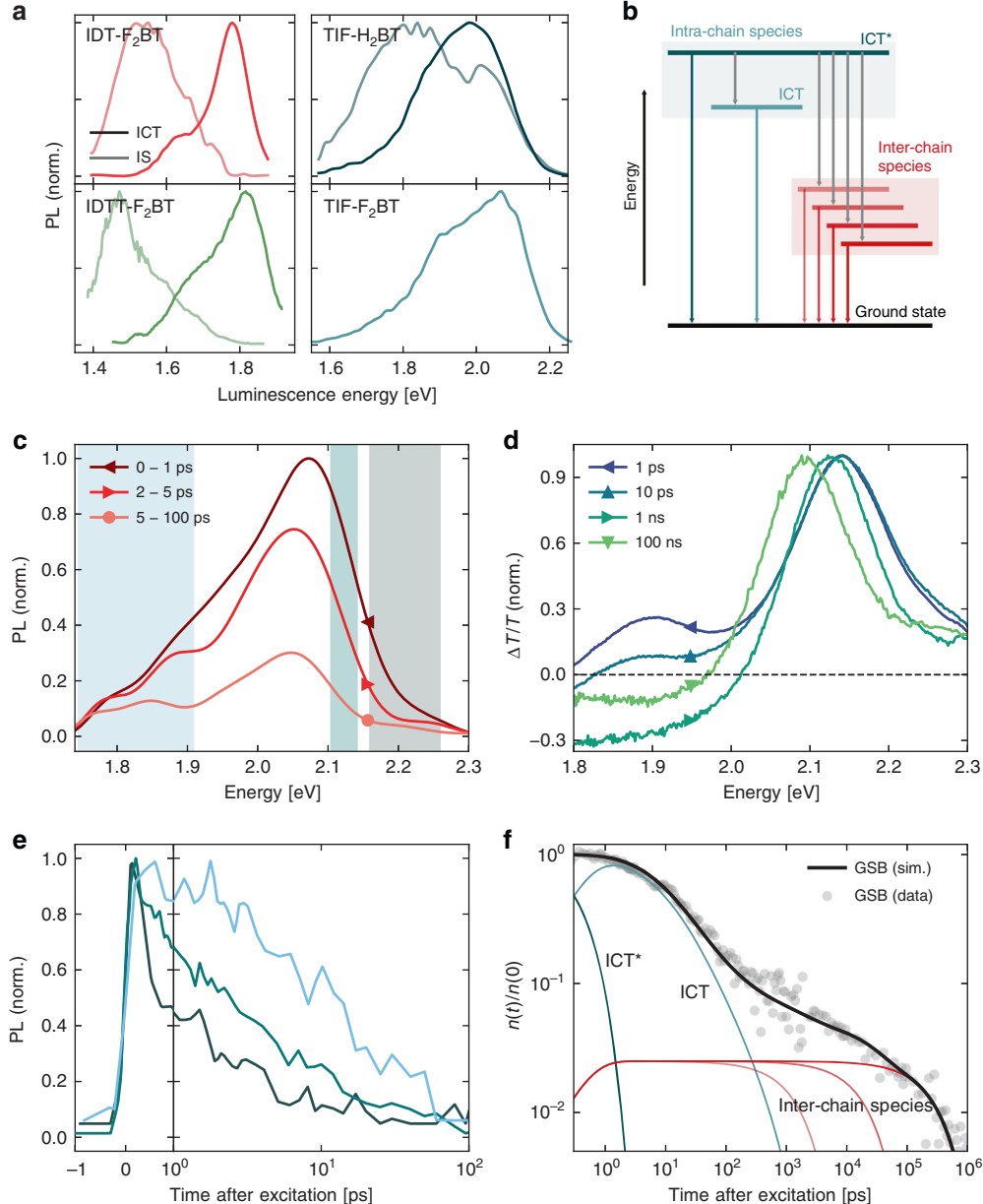

**Fig. 3** Transient optical characterisation. **a** PL spectra of the prompt and delayed emissive species extracted from the genetic algorithm. **b** Energy level diagram summarising the exciton dynamics. For TIF-H$_2$BT: (**c**) ultrafast PL spectra normalised to the maximum of the first time-slice shown; (**d**) normalised TA spectra for different time delays; (**e**) kinetics of the PL decay in the different energy bands shaded in (**a**); and (**f**) GSB with population fits normalised to the maximum signal together with a fit of the contributions from different excited state species according to the kinetic model

(Supplementary Note 4). In the spectra, there are three spectro-scopic features: a ground-state bleach (GSB) above 2.0 eV, stimulated emission (SE) below 2.2 eV, and a broad photoinduced absorption over the entire probe band but increasing at low energies (PIA). In brief, on the timescales of ICT luminescence, the signal is composed of all three spectroscopic features. By ~2 ns, the SE decays to zero, leaving a GSB that persists to ~µs, and a PIA continues to evolve over this time. While there is SE, the exciton population is mostly ICT excitons. However, when SE is depleted, only interchain species (which have a weak SE) are left, leading to the persisting GSB. For this reason, in systems where lifetime $\tau_{IS}$ exceeds $\tau_{ICT}$ by more than ~2×, the approximate yield for forming interchain species ($\phi_{IS}$) can be estimated by the fraction of GSB remaining after SE has finished (although a more accurate extraction methodology is ultimately used, and discussed

in Supplementary Note 4). In addition to the luminescent interchain species, there exist several additional non-emissive aggregate states that recombine on longer timescales. These have different absorption spectra and leading to the PIA evolution long after $\tau_{IS}$. This agrees well with the large bandwidth of sub-$E_g$ states in the absorption spectra. This kinetic picture for TIF-H$_2$BT is summarised in Fig. 3f, with the general states for this polymer family in Fig. 3b.

A full discussion of the assignment of species for IDT-H$_2$BT and TIF-H$_2$BT is provided in Supplementary Note 4 (Supplementary Figs. 7–9 comprising ultrafast spectra and kinetics, and Supplementary Figs. 10 and 11 comparing ultrafast and nanosecond TA spectra of all the materials in this study). We also summarise the complete exciton dynamics in Fig. 3b and Supplementary Note 4.

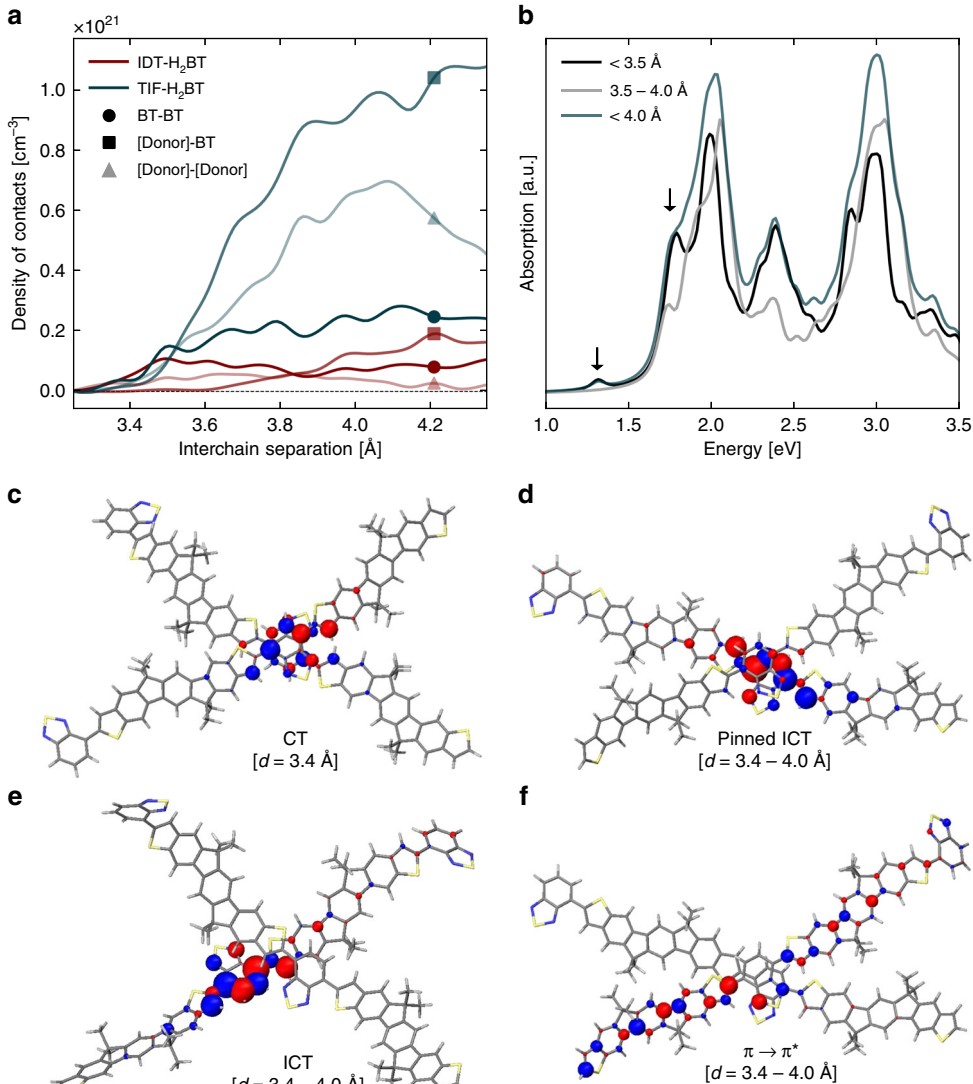

**Fig. 4** Theoretical calculations of excited states at close contact points. **a** Radial distribution functions for various interchain coupling motifs for IDT-H$_2$BT and TIF-H$_2$BT. **b** Absorption intensity for the closest interchain coupling motif for TIF-H$_2$BT, shown below. For various model geometries for the close crossing of two polymer chains, transition densities for absorption features are shown: the CT formed at an interchain separation of 3.4 Å (**c**), and the pinned ICT (**d**), ICT (**e**), and $\pi \to \pi^*$ (**f**) transitions at various geometries in the range 3.4–4.0 Å

**Modelling interchain interactions**. To better understand the nature of the aggregate and interchain species excited states (identified above), we have performed molecular dynamics (MD) and quantum-chemical (INDO/SCI) excited-state calculations. From MD simulations of IDT-H$_2$BT and TIF-H$_2$BT amorphous films, we have assessed the nature and density of the interchain contacts. In each case, the sterical bulkiness (of the solubilising sidechains) in the immediate vicinity of the central six-membered ring hinders close and extended interchain interactions in cofacial-like arrangements. Instead, interchain contacts only occur through local crossing points between polymer chains that are rotated about their out-of-plane axis. This sterical effect is pronounced for both polymers, and the closest interchain separations ($d$) of ~3.3 Å are therefore realised via the rotated packing of BTs. This is shown in Fig. 4a (and the methodology is given in Supplementary Note 5, Supplementary Fig. 16) where the density of interchain contacts via BT-BT interactions (circles) increases uniformly for the two polymers (from zero for $d < 3.3$ Å to a constant value for $d > 3.6$ Å). The high device mobility of IDT-H$_2$BT is likely to rely in part on BT-BT close-crossing

bridges between adjacent chains in addition to the stiffness of the conjugated backbones.

The difference between the two polymers is most visible when looking at the close contact points involving the donor segments. The density of interchain contacts for [Donor]-BT and [Donor]–[Donor] interactions are shown in Fig. 4a as squares and triangles, respectively. Here, the density histograms increase more quickly above 3.5 Å for TIF-H$_2$BT than for IDT-H$_2$BT, so it is clear that polymer chains form a higher density of donor-mediated close-contacts via the TIF donor subunit rather than IDT. This is achieved by extending the length of the backbone, which reduces the steric hindrance conferred by the sidechains. In fact, this design strategy for TIF-H$_2$BT increases the density of all three (donor–donor, donor–acceptor, and acceptor–acceptor) interchain interactions compared with IDT-H$_2$BT, and is likely to be the origin of the improved mobility of TIF-H$_2$BT over IDT-H$_2$BT.

From our MD simulations of the bulk materials, we selected various TIF-H$_2$BT chain pairs with close-contact points (of different separations) and performed excited-state calculations

using the semi-empirical INDO/SCI approach discussed in the methods section. In Fig. 4b, we show representative optical absorption spectra calculated for these TIF-H$_2$BT aggregates with among the closest interchain separations of $d = 3.4$ Å (black) and a more typical close-crossing point with $d = 3.4 - 4.0$ Å (grey) to illustrate the influence of increased interchain coupling on the optical properties. We observe sub-energy gap transitions at 1.32 and 1.75 eV (labelled with arrows), and this is in excellent agreement with the experimental spectra, confirming our earlier assignment of these absorption features being induced by interchain interactions. In particular, the calculations suggest that the lowest-lying electronic excitations pertain to spatial regions of the films with the shortest interchain separations. Since the polymers in this study exhibit a high degree of structural disorder, we expect a distribution of interchain separations, and a broadening of these transitions in thin films. To a good approximation, the sum of these extreme cases (blue spectrum in Fig. 4b) reproduces the main absorption features observed in the experimental spectrum in Fig. 2e, including the sub-$E_g$ features.

To understand the difference between the two sub-$E_g$ features predicted theoretically, we calculate the transition density distributions for the selected cases in Fig. 4c–f. While ICT (Fig. 4e) and $\pi \rightarrow \pi^*$ (Fig. 4f) transitions (at 2.0 and 3.1 eV, respectively, calculated for an interchain separation of $d \sim 4.0$ Å) extend over large portions of the backbone, the sub-$E_g$ features at 1.31 eV (Fig. 4c) and 1.75 eV (Fig. 4d) are more localised. Characterised by a significant interchain delocalisation over the two chains, the lowest energy absorption at 1.40 eV has a pronounced charge transfer (CT) character; it is mediated predominantly by strong coupling of the BT units on adjacent chains. The origin of the feature at 1.75 eV is different, namely the degree of interchain charge transfer is small. It is more similar to the ICT, except that it is stabilised by the presence of the second chain, and is therefore 'pinned' to the region of the close-contact point. This pinned ICT (pICT) has a stronger oscillator strength than the CT, and is therefore more likely to be luminescent. We identify the observed emissive IS discussed above with emission from pICT states. Broadly speaking, the pICT gains radiative intensity at the expense of the ICT, as the density of close-contact points increases, though the absolute oscillator strengths of course also depend on the detailed local geometry at those crossings. Since this sub-$E_g$ feature is caused directly from the close interchain contacts, and modulates its oscillator strength according to the interchain geometry, efficient luminescence from the pICT with strong oscillator strength provides a spectroscopic probe of the close-crossing points in the polymer film.

In these polymers, decreasing steric hindrance by backbone elongation allows the formation of a greater density of close-crossing points, which can be probed in amorphous systems using the pICT luminescence. This leads to the formation of luminescent interchain species that contribute to the luminescence quantum efficiency in films, while triggering interchain percolation pathways for charge carrier transport.

Since we observe that luminescence from the pinned ICT in IDT-H$_2$BT is weak compared with TIF-H$_2$BT, we posit that this arises from different densities of close-contact points in the two copolymers. To confirm this statement, we complete the photophysical picture, and use interchain species formation efficiencies and the luminescence rates for the polymers to finally unpack the nature of recombination in this family of polymers.

The total luminescence quantum yield $\Phi$ can be expressed in terms of the quantum efficiencies of energy transfer ($\phi_{i \rightarrow j}$ from excited state $i$ to another excited state $j$) and PL ($\phi_{i \rightarrow 0}$ from state $i$ to the ground state). In the limit that $\phi_{ICT^* \rightarrow 0}$ and $\phi_{ICT \rightarrow IS}$ are

small, $\phi_{ICT^* \rightarrow IS} \rightarrow \phi_{IS}$ and we obtain (neglecting up-conversion),

$$\Phi = (1 - \phi_{IS}) \cdot \phi_{ICT \rightarrow 0} + \phi_{IS} \cdot \phi_{IS \rightarrow 0} \qquad (2)$$

Furthermore, the ratio of $\phi_{i \rightarrow 0}$ for $i$ luminescent states can be expressed in terms of the spectrally integrated PL intensities ($I_i$) assuming first-order dynamics

$$\frac{\phi_{ICT \rightarrow 0}}{\phi_{IS \rightarrow 0}} = \frac{I_{ICT}}{I_{IS}} \cdot \frac{\phi_{IS}}{1 - \phi_{IS}} \qquad (3)$$

Therefore, independent measurement of $I_i$ and $\Phi$ from PL spectra and decay kinetics, and $\phi_{IS}$ from TA spectroscopy, allows for the extraction of the PLQEs ($\phi_{PL}$) of both the intra-chain ICT and interchain luminescence pathways. Full details of the methodology are given in Supplementary Note 4 (Supplementary Figs. 12–15 and values in Supplementary Tables 3–5). Since we are unable to distinguish between luminescent and 'dark' interchain species (i.e., pICT and CTs/PPs) in TA, our treatment considers them together. Because of this, hereunder, we report the $\phi_{PL}$ of IS as a whole, which can be thought of as the lower bound on the pICT $\phi_{PL}$.

We show the dependency of $\phi_{PL}$ for the (intra-chain) ICT and interchain species on the energy gap of the transition (determined by the energy of the PL spectral maximum) in Fig. 5. The trend is clear, and for the ICT, the $\phi_{PL}$ increases with energy gap, consistent with the energy gap law. This trend also holds for interchain species, although IS $\phi_{PL}$ follows a different quasi-exponential relation with the transition energy gap to the ICT. Interestingly, IS (and therefore pICTs) have a higher $\phi_{PL}$ than the ICT at the same transition energy gap. This can be seen by comparing $\phi_{PL} \sim 0.02$ for the ICT (closed circles) of the polymers with luminscence at ~1.8 eV, with $\phi_{PL} \sim 0.12$ for the IS (open circles) of TIF-H$_2$BT and TIF-F$_2$BT, whose luminescence is centred at the same energy. The act of ICT pinning leads to a redshifted emission compared with the isolated chain, but this occurs without substantially compromising $\phi_{PL}$. This is profound, since it presents a strategy to increase the performance of red and NIR polymer emitters by encouraging specific interchain interactions instead of limiting them.

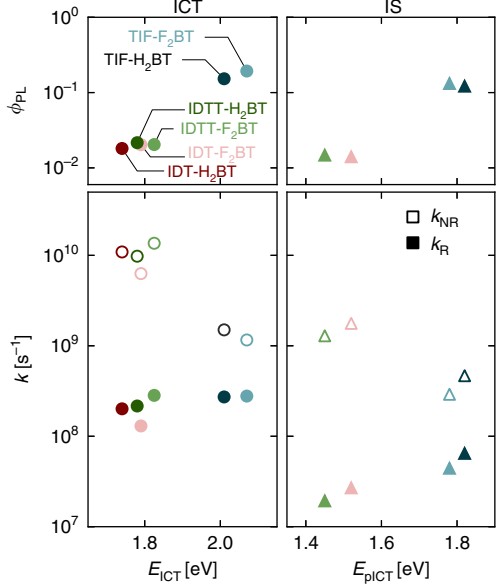

**Fig. 5** Extracted recombination parameters. Intra-chain ICT (circles) and interchain pICT (triangles) excitons parametrised in terms of the luminescence energy $E$: PLQE of the transition, rates of recombination, colour-coded by polymer

To understand this insight further, we determine the radiative ($k_R$) and non-radiative ($k_{NR}$) rates of exciton recombination using knowledge of $\phi_{PL}$ and $\tau_{PL}$, and precluding further energy transfer to other states. We plot $k_{NR}$ against the luminescence energy in Fig. 5. In intra-chain ICTs (open circles), $k_{NR}$ is generally high, particularly for transitions below 1.9 eV, but decreases by an order of magnitude for transitions at ~2.0 eV. This behaviour is predicted by the energy gap law, as strong wavefunction overlap with the ground state at low energy gaps manifests itself as a high $k_{NR}$. By comparison, $k_{NR}$ values for the pICT emission (open triangles) follow the same trend, and are uniformly slower than their ICT counterparts, even at an equivalent transition energy. Importantly, $k_{NR}$ in pICT excitons is significantly slower at an equivalent transition energy. This indicates a strong dependency of the non-radiative recombination rate on the exciton geometry, and shows that pICT excitons, which are likely much less mobile, are thus less prone to quenching at defective sites. This assertion is also supported by the smaller decrease in $k_{NR}$ for IS with increasing energy than for the ICTs.

We also show $k_R$ in Fig. 5. For ICTs (filled circles), $k_R$ is nearly independent of transition energy. The radiative rate is fast, and approximately constant at ~$10^8 \, s^{-1}$. For excitons, $k_R$ arises directly from the degree of overlap of hole and electron wavefunction. In this family, a uniform $k_R$ implies wavefunctions overlap to a similar degree. Indeed, we find from DFT that the calculated HOMO and LUMO wavefunctions in these donor–acceptor polymers have the same geometric motif: the HOMO is more delocalised over the polymer backbone, while the LUMO wavefunction is localised on the BT acceptor unit.

For the pICT emission (coloured triangles), $k_R$ is determined by the same physics. In fact, our calculations show that both ICT and pICT source their absorption/emission cross-section from the overlap between hole and electron density alternating over the donor and acceptor units along the same polymer chains. Figure 4b shows that short interchain contacts steer pICT with sizeable oscillator strength. While $k_R$ is still lower than for ICTs, we measured a radiative rate for pICTs in TIF-H$_2$BT which is substantially higher than for IDT-F$_2$BT and IDTT-F$_2$BT, and higher even than TIF-F$_2$BT.

We can therefore use $k_R$ of the pICT transition as a powerful probe for the degree of spatial coupling at close-crossing points. Since close contacts trigger luminescent pICTs, our treatment of such species is most sensitive to points in the polymer microstructure which are likely to mediate charge transport between chains. Our results demonstrate that TIF-H$_2$BT forms a greater density of such close-crossing points, more than the other polymers in this study. This is likely to be the origin of the superior $\mu$ observed in TIF-H$_2$BT over IDT-H$_2$BT and IDTT-H$_2$BT. Furthermore, towards the further improvement of near-amorphous donor-*alt*-acceptor polymers, we suggest that increased interchain interaction can be achieved via the following design rules: increase the length of the backbone (as we did for IDT-H$_2$BT to make TIF-H$_2$BT), decrease the push-pull polarisation (as can be seen from the comparison between TIF-H$_2$BT and TIF-F$_2$BT) and minimise sidechain sterical hindrance (as determined by the lower $k_R$ of IDTT-F$_2$BT with its bulkier side chains).

While a significant fraction (~20%) of the PL originates from the interchain (pICT) pathway, the high $\Phi$ in TIF-H$_2$BT arises mainly from the high $\phi_{PL} > 0.16$ of the ICT. Since ICT luminescence efficiency is determined by the suppression of $k_{NR}$ rather than increased $k_R$, its high $\Phi$ is primarily owing to the energy gap law with $E_g \sim 2.0$ eV. The increased $E_g$ is achieved by increasing the electron density on the donor, which decreases the overall polarisation of the polymer repeat unit, and can be achieved by design rules above. Concomitantly with the final design rule, the elongated repeat unit allows for a higher degree of wavefunction overlap at close-crossing points between chains, giving an improved contribution to the luminescence from pICT species, which benefits the mobility at the same time. Ultimately, for a system with interchain pICT luminescence (with $\phi_{PL} \sim 0.12$), increasing $\phi_{IS}$ decreases $\Phi$ marginally, but at the same time provides an opportunity to access strongly redshifted luminescence (by >200 meV) at a luminescence efficiency close that of a bluer ICT, an improvement of more than 10× over materials whose ICTs luminescence at the same energy. We have demonstrated that this redshifted pathway can also be accessed with electrical injection in an OLED architecture (Supplementary Fig. 4). Combined with its high $\mu$, this represents an important potential path towards the fabrication of efficient red and NIR polymer OLEDs, towards electrically-driven lasing in these systems, which are currently hampered by the former trade-off between $\Phi$ and $\mu$. Notwithstanding, a near-amorphous TIF-H$_2$BT benefits from a generally low $\phi_{IS}$ compared with $\phi_{IS} > 0.3$ in lamellae-forming semicrystalline polymers[14]. This allows for the simultaneous improvement of $\Phi$ and $\mu$, since aggregation-induced non-radiative effects are limited in the absence of long-range crystalline order and a high $\phi_{IS}$.

## Discussion

In the present work, we have demonstrated a class of conjugated polymers with low degree of energetic disorder, that exhibit simultaneously a high carrier mobility >2 cm$^2$/Vs and a high photoluminescence quantum efficiency >15% and, to the best of our knowledge, the highest $\Phi \cdot \mu$ values reported thus far for conjugated polymers, which outcompete the state-of-the-art by >10×. The high $\Phi \cdot \mu$ product obtained in polymers such as TIF-H$_2$BT is the combined result of a higher energy gap induced by extended conjugated units and an increased number of close contact points that mediate the 3D charge-percolation pathway in devices. This provides a clear strategy for designing a new generation of highly luminescent conjugated polymers that no longer suffer from low carrier mobilities, and could enable a new generation of optoelectronic, light-emitting and photovoltaic devices with higher performance and simpler device architectures, and potentially even electrically-driven organic lasers. We have also presented a careful spectroscopic and theoretical study of their photophysics and have shown that the close-crossing points in the polymer network, that are critical for achieving high carrier mobilities, are associated with a long-lived, red-shifted emission due to pinned ICT states. These do not only provide a powerful spectroscopic probe of the degree of interchain interactions at the close-crossing points, but also provide an interesting potential route to more efficient NIR emissive devices, as the pICT states are found to be more robust against non-radiative recombination processes than ICT states at equivalent energies.

## Methods

**OFET fabrication and characterisation**. Interdigitated bottom-contacts (Au, 20 nm, $L = 20 \, \mu m$, $W = 1$ mm) were patterned using photolithography on smoothed alkali glass. After solvent sonication and plasma-treatment, a film (~20 nm) was cast from a precursor solution (10 g/L, *o*-DCB). A Cytop (CTL-809M, Asahi Glass Co.) film (~500 nm) was spin-coated and annealed (90°, 5–20 mins). All solution processing took place under a N$_2$-atmosphere. A top-gate was thermally evaporated (Ag, 20–30 nm) under high-vacuum. The device was measured (4155B SPA, Agilent) under a N$_2$-atmosphere using bespoke LabView (National Instruments) software.

**Steady-state absorption**. Thin polymer films were spin-coated as for OFETs. Time-integrated absorbance was measured ($T = 300$ K) with a HP8453 spectrophotometer, (Hewlett-Packard) between 250–900 nm.

**Photo-thermal deflection spectroscopy (PDS)**. Our home-built PDS setup uses a monochromatic pump beam which is directed on the sample (film on Spectrosil substrate). Absorption produces a thermal gradient near the sample surface via non-radiative relaxation induced heating. This results in a refractive index gradient in the area surrounding the sample surface. This refractive index gradient is further enhanced by immersing the sample in an inert liquid FC-72 Fluorinert (3M Company) which has a high refractive index change per unit change in temperature. A fixed wavelength CW laser probe beam is passed through this refractive index gradient producing a deflection proportional to the absorbed light at that particular wavelength, which is detected by a photo-diode and lock-in amplifier. Scanning through different pump wavelengths builds up the absorption spectra.

**Ultra-fast transient aborption**. The output of a Ti:Sapphire amplifier system (Spectra-Physics Solstice Ace) operating at 1 kHz and generating 90 fs pulses was split into pump and probe beam paths. The visible and near-infrared broadband probe beams were generated in home-built noncollinear optical parametric amplifiers. The transmitted pulses were collected with an InGaAs dual-line array detector (Hamamatsu G11608-512) driven and read out by a custom-built board (Stresing Entwicklungsbüro). ps-measurements were carried out with a narrow-band (10 nm FWHM) pump beam provided by a TOPAS optical parametric amplifier (Light Conversion). ns-measurements were carried out with a 355 nm narrowband pump beam provided by a Q-switched Nd:YVO4 laser (Advanced Optical Technologies).

**Steady-state PL**. Samples were exposed to a pulsed (407 nm, 4.1 μJ/cm$^2$, 300 ps) diode laser (PDL 800-B (trigger), LDH-P-C 400B (laser head), PicoQuant) and PL was integrated (SpectraPro 2500i, Acton Research Co.) over seconds under a N$_2$ atmosphere.

**Time-correlated single photon counting**. With the same setup above, photoluminescence decays were determined using time-correlated single photon counting. Detection at a specific wavelength was achieved using a monochromator, and used a PMT (LifeSpec-ps, *Edinburgh Instruments*).

**PLQE**. Encapsulated films were exposed to a CW (405 nm, ~1 mW) diode laser inside an integrating sphere (6 inch, LabSphere). The integrated signal was coupled to a PL spectrometer (SR-303i-B, Andor Technology) which had a Si detector (350–1050 nm). The intensity of laser signal was quantified both with and without a loaded sample.

**Ultrafast transient grating PL**. We use Ti:Sapphire amplifier system (Spectra-Physics Solstice) operating at 1 kHz generating 90 fs pulses was split into the pump and probe beam arms. The pump beam was generated by second harmonic generation (SHG) in a BBO crystal and focused onto the sample. PL is collimated using a silver off-axis parabolic mirror and focused onto the gate medium. Approximately 80 μJ per pulse of the 800 nm laser fundamental is used for the gate beams, which is first raised 25 mm above the plane of the PL to produce a boxcar geometry and split into a pair of gate beams using a 50:50 beam splitter. The gate beams are focused onto the gate medium (fused silica), crossing at an angle of ~5° and overlapping with the focused PL. The two gate beams interfere and create a transient grating in the gate medium due to a modulation of the refractive index via the optical Kerr effect. Temporal overlap between the two gate beams is achieved via a manual delay stage. The PL is then deflected on the transient grating causing a spatial separation of the gated signal from the PL background. Two lenses collimate and focus the gated signal onto the spectrometer entrance (Princeton Instruments SP 2150) after long-pass and short-pass filters remove scattered pump and gate light, respectively. Gated PL spectra are measured using an intensified CCD camera (Princeton Instruments, PIMAX4). The (~10 ns) electronic shutter of the intensified CCD camera was used to further suppress long-lived PL background. PL spectra at each gate time delay are acquired from ~10$^5$ laser shots. The time delay between pump and gate beams is controlled via a motorised optical delay line on the excitation beam path and a LabVIEW data acquisition program.

**Theoretical calculations**. To calculate structural and optical properties of TIF-H$_2$BT films, we first simulated the supramolecular organisation of an amorphous phase following the protocol described previously for IDT-H$_2$BT[27]. This is discussed in more detail in Supplementary Note 5. To characterise the distance between the chains or their subunits (TIF and/or H$_2$BT), the radial distribution functions between the different subunits have been built from the 400 snapshots of the analysis molecular dynamics (MD) trajectory, taking as reference points either the centre of the central bond for H$_2$BT, or the centre between two characteristic atoms of the corresponding rings for IDT (see Supplementary Fig. 16).

To probe the optical properties of TIF-H$_2$BT chains, we performed quantum-chemical calculations using the semiempirical intermediate neglect of differential overlap (INDO) Hamiltonian[59], combined with single configuration interaction (SCI). These calculations were performed for individual or interacting pairs of TIF-H$_2$BT oligomers ($n = 2$) extracted along the 2 ns-long MD run. In each case, the configuration space involves single excitations between the 25 highest occupied

MOs and the 25 lowest unoccupied MOs. In addition to computing excitation energies and oscillator strengths, we assessed the nature and localisation of the electronic excitations contributing to all relevant optical absorption bands through an analysis of the corresponding atomic transition density distributions, which provide a local mapping of the electronic excited-state wavefunctions. The analysis was run for pairs of neighbouring chains extracted from the MD trajectory, considering as a selection criterion a centre-to-centre distance smaller than 4 Å.

## Data availability
The data presented in the paper will be made available after acceptance of the paper on the University of Cambridge's Apollo Open Data Repository.

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

## Acknowledgements

We thank the Engineering and Physical Sciences Research Council (EPSRC) for funding through a programme grant (EP/M005143/1). T.H.T. thanks EPSRC for an Industrial CASE studentship, as well as the Cambridge Commonwealth Trust for funding. D.H. would like to thank the Doctoral Training Centre in Plastic Electronics EP/G037515/1, the Worshipful Company of Armourers and Brasiers, and St. Edmunds College, Cambridge. M.N. acknowledges financial support from the European Commission through a Marie-Curie Individual Fellowship (747461). We thank Aurélie Morley and Merck Chemicals Ltd. for providing IDT-F2BT, IDTT-H2BT and IDTT-F2BT. The work in Mons was supported by the Belgian National Science Foundation, F.R.S.-FNRS and by the European Commission/Région Wallonne (FEDER–BIORGEL project). Computational resources have been provided by the Consortium des Équipements de Calcul Intensif (CÉCI), funded by F.R.S.-FNRS under Grant No. 2.5020.11, as well as the Tier-1 supercomputer of the Fédération Wallonie-Bruxelles, infrastructure funded by the Walloon Region under the grant agreement n1117545. The research in Mons is also through the European Union's Horizon 2020 research and innovation program under Grant Agreement No. 646176 (EXTMOS project). D.B. is a FNRS Research Director.

## Author contributions

T.H.T. and M.N. fabricated and measured the FETs, and T.H.T., D.J.H. and J.M.R. measured the PLQEs and PL lifetimes. T.H.T. and A.J.G. measured the TA spectroscopy, A.S. measured the PDS spectra, and J.A. performed the charge accumulation spectroscopy. H.C. and I.M. synthesised IDT-H2BT, TIF-H2BT and TIF-F2BT. V.L., Y.O. and D. B. performed the theoretical calculations. T.H.T. and D.J.H. wrote the manuscript with H.S., D.B., S.M.M. and all authors were involved in revising the manuscript.
