## [Peer Review File · Nature Communications]

Reviewers' comments:

Reviewer #1 (Remarks to the Author):

In this manuscript NCOMMS-18-29533, Thomas and co-authors propose the characterization of conjugated copolymers with both high carrier mobility and luminescence quantum yield. The chosen topic is meaningful and the result presents a critical issue in organic optoelectronics. This is due to the scarcity of copolymers simultaneously satisfy high mobility and electroluminescent efficiency. The data present by the authors are convincing, however, the physics reasoning and narration behind the devices require re-consideration and clarification. Therefore, I suggest this work publishable in Nature Communications after following revisions.

1. Page 2. The authors claim DPP and PBTTT don't show a direct relationship between Bandgap and luminescence quantum efficiency due to charge injection or trapping. The fact is IDT- and TIF-series materials show lower HOMO levels and more likely to hinder hole injection. Also, the mobility dependence on gate voltage from DPP polymers can be suppressed using Cytop, as described by Yun-Hi Kim (J. Am. Chem. Soc., 2013, 135 (40), pp 14896–14899). I prefer to attribute mobility dependence on gate voltage to polymers' feature.

2. F8T2 is known for its light-emitting OFET, which has been published by the same group (from the authors). Photophysics (PDS and PL) spectra present here can indicate the used polymers' potential as light-emitting devices. However, I recommend the authors to compare these polymers' capability more directly, by fabricating light-emitting transistors and conventional device characterization.

3. In Figure 3e–f, interchain species emits photons in the low energy region and can serve as NIR LED as claimed by the authors. I suggest the authors compare the NIR light-emitting efficiency from TIF polymers with traditional low bandgap copolymers to show whether TIF is more advantageous in this aspect.

Reviewer #2 (Remarks to the Author):

In the present manuscript NCOMMS-18-29533, the authors discuss a set of polymers that show both high carrier mobility (here holes) and luminescence quantum yield. The motivation of this work is the understanding of principles that lead to highly luminescent materials with high carrier transport mobilities that will ultimately allow for higher performance optoelectronic devices such as electrically pumped lasers. The work is very interesting and extensive. It is a good fit for the scope of Nature Communications. However, the manuscript has some shortcomings which make a major revision mandatory. Please find below my points:

1. The manuscript contains some typos: E.g. Page 8 'JDOS' Is it really that - if so, it is only vaguely defined as joint (J) density of states (DOS) once, or 'DOS'?, page 8 'equilibrated' - please check the manuscript carefully.

2. Determination of PLQE: The authors state that the PLQE is determined for encapsulated samples. How is the encapsulation made? Typically, the Encapsulation induces significant absorption by the glue used that hampers the PLQE values.

3. Figure 4: The figure caption is missing information for panels d, e, and f. Furthermore, why is the distance for the panels c and d given, but not for e and f. And, very important, the figure caption does not include the description what the panels c-f are showing. Please add the information that there are each two polymer chains crossing. Otherwise, the reader has a hard time to figure this out. // Also, how are the distributions determined. Add information to the caption.

4. Figure 1: There are two problems here: a) The materials part of this study (colored dots) are

not labelled, so there is no way to find out, which data point corresponds to which actual material.
b) The literature values are all labeled with a grey triangle. While this may be done to give the overall trend, it is important for the reader to know the origin of each data set. Either include this into this figure, or add an SI plot, disclosing all literature sources.

5. On page 9, the authors state: 'Separately, we have observe similar dynamics for IDT-H2BT in unpublished work.' This is unsatisfying for the reader. Can't the essence of this work be included in the SI? A high level publication such as in Nature Communications should allow this extra detail.

6. Figure 5: Again, no label to the actual molecules. Please include proper legends.

7. Figure 5: For the ϕ_{PL} , it would be better to have the y-axis scaled all the way to 1 (unity), as this represents the upper limit. And further, put both k_r and k_{nr} axis to the same ranges, to better compare.

Reviewer #3 (Remarks to the Author):

In this manuscript, the authors reported a conjugated polymer of both high hole mobility and fluorescence quantum efficiency, which may foresee applications in electrically excited polymer lasers. Comparative study of a series of conjugated polymers, including model polymer IDT-H2BT, backbone fluorinated and pi-elongated derivatives, has demonstrated that interchain "close-crossing points" not only facilitate percolation of charge carries in the polymer network, but also provide emissive species of long lifetime. Finally, the highest $\Phi \cdot \mu$ value was obtained from TIF-H2BT via proper interplay between mobility and luminescence. This is basically an interesting and enlightening research considering the theoretical and experimental characterizations, I would recommend publication of this manuscript but after minor revision by the authors by addressing the following points:

1. The contact resistance was mentioned across the electrical characterization section, however, experimental data of contact resistance were not included and should be given. As inferred by the output characteristics, the difference between the contact resistance of TIF-H2BT and IDT-H2BT was surprisingly small as for ionization potential 5.7 eV of TIF-H2BT (Adv. Mater. 2017, 29, 1702523) vs 5.4 eV of IDT-H2BT (J. Am. Chem. Soc., 2010, 132 (33), pp 11437–11439), and surprisingly large as for IP 5.7 eV of TIF-H2BT vs 5.8 eV of TIF-F2BT (Adv. Mater. 2017, 29, 1702523), given Au contacts were not modified to improve injection.
2. As shown in Figure S1, molecular weights and PDIs varied within this series of polymers. It should be discussed whether the electrical and photophysical properties are dependent on these two factors.

Reply to Reviewers' comments:

We thank all three reviewers for the generally positive evaluation and the helpful and constructive suggestions. We respond below to each of the points raised.

Reviewer #1 (Remarks to the Author):

In this manuscript NCOMMS-18-29533, Thomas and co-authors propose the characterization of conjugated copolymers with both high carrier mobility and luminescence quantum yield. The chosen topic is meaningful and the result presents a critical issue in organic optoelectronics. This is due to the scarcity of copolymers simultaneously satisfy high mobility and electroluminescent efficiency. The data present by the authors are convincing, however, the physics reasoning and narration behind the devices require re-consideration and clarification. Therefore, I suggest this work publishable in Nature Communications after following revisions.

We thank the reviewer for their positive review, and agree with most of the points made below.

1. Page 2. The authors claim DPP and PBTTT don't show a direct relationship between Bandgap and luminescence quantum efficiency due to charge injection or trapping. The fact is IDT- and TIF-series materials show lower HOMO levels and more likely to hinder hole injection. Also, the mobility dependence on gate voltage from DPP polymers can be suppressed using Cytop, as described by Yun-Hi Kim (J. Am. Chem. Soc., 2013, 135 (40), pp 14896–14899). I prefer to attribute mobility dependence on gate voltage to polymers' feature.

We believe this might be due to a misunderstanding: The reviewer is referring to the statement “This does not reflect a direct relationship between [mobility] and [energy gap]”, since we *do* believe that energy gap (or bandgap) is directly related to luminescence quantum efficiency in thin films; indeed, we state this multiple times in the manuscript, and also show this in Figure 1a (top). The phenomenon of the energy gap law is well-documented in the literature (Englman, Jortner. *Molecular Physics* **18**, 2, 145 (1970), Caspar, Meyer. *Journal of Physical Chemistry*, **87**, 6, 952). The statement refers to the apparent relationship between *mobility* and energy gap, which we believe is not direct, i.e. it is not due to some intrinsic reason that would cause high mobility materials to have a low energy gap, but is rather due to the difficulties of injecting charges and avoiding extrinsic trapping in materials with larger energy gap and deeper HOMO levels.

We agree that OFET devices using polymers with deeper HOMO levels often suffer from a gate-voltage dependence of the mobility, and that this effect can be minimised by optimising the solution processing of the devices. We discuss this in the context of our materials in the supplementary information (Section S2), alongside a detailed discussion of how to ensure a robust mobility extraction in spite of this. We thank the reviewer for raising this point and we have clarified this in the revised manuscript.

2. F8T2 is known for its light-emitting OFET, which has been published by the same group (from the authors). Photophysics (PDS and PL) spectra present here can indicate the used polymers' potential as light-emitting devices. However, I recommend the authors to compare these polymers' capability more directly, by fabricating light-emitting transistors and conventional device characterization.

We thank the reviewer for their comment; and are pleased that they agree steady-state optical spectroscopy (including PDS and PL) provides compelling evidence for polymers utility as light-emitters in various architectures. In this work, our focus was to provide detailed experimental evidence that mobility and a high quantum yield can be achieved together, both of which being conferred by close interchain contacts. The discovery of a family of polymers that behave in this way provides important experimental evidence contrary to the common opinion that there is always a trade-off between these two important figures of merit. We agree that it is important to show that this family of polymers can be performant in electroluminescent devices, and for this we direct the reviewer to Figure S3 and to a study by some of us (Harkin et al., *Advanced Materials*, **28**, 30, 6378 (2016)) which shows the efficacy of these materials in OLED architectures. Regarding their performance in OLEFETs, and a conventional device characterisation, we do not believe this aligns with the scope of this manuscript; and that a pursuant study detailing the device optimisation of OLEFETs, one which enlists the design rules explored in this manuscript, is required. To fabricate OLEFETs with optimum performance requires significant effort in device optimisation, in

particular of electron and hole injection at source and drain contacts, respectively, as well as light outcoupling. This goes beyond the scope of the present paper which is focussed on the investigation of the correlation between charge transport and photophysics.

3. In Figure 3e–f, interchain species emits photons in the low energy region and can serve as NIR LED as claimed by the authors. I suggest the authors compare the NIR light-emitting efficiency from TIF polymers with traditional low bandgap copolymers to show whether TIF is more advantageous in this aspect.

The reviewer raises an interesting point, and we thank them for the opportunity to clarify this. He/she asks for a comparison of the NIR light-emitting efficiency from TIF polymers with traditional low energy-gap polymers. Indeed, this is achieved by comparing Figure 5 (top right) and Figure 1a (top). This comparison is discussed in the main text (end of page 14), where we show that the PL quantum efficiency of the interchain states (IS) is higher by more than 5x compared to the internal charge transfer states (ICT) which luminesce at the same energy. It is unusual to find a polymer whose ICT luminescence is near 1.8 eV and whose PL quantum efficiency exceeds 10%; and we attribute this to the slow non-radiative recombination rate of the pICT.

We agree that further work involving a detailed study of light-emitting devices is the next step towards realising high efficiencies in the red-NIR band. However, due to the similar origin of the two emissive species, we find that the luminescence (both PL and electroluminescence) is strongly coupled (Figure S3, Figure 3c). In addition, the broad JDOS of the pICT leads to a broad sub-energy gap luminescence spectra in TIF-H2BT and TIF-F2BT, overlapping considerably with the transition at the energy-gap edge (Figure 3a). Due to these complexities, it seems not appropriate to us to make specific claims to whether the TIF polymers could be used for efficient NIR LEDs, this needs to be investigated in future work.

Reviewer #2 (Remarks to the Author):

In the present manuscript NCOMMS-18-29533, the authors discuss a set of polymers that show both high carrier mobility (here holes) and luminescence quantum yield. The motivation of this work is the understanding of principles that lead to highly luminescent materials with high carrier transport mobilities that will ultimately allow for higher performance optoelectronic devices such as electrically pumped lasers. The work is very interesting and extensive. It is a good fit for the scope of Nature Communications. However, the manuscript has some shortcomings which make a major revision mandatory. Please find below my points:

We thank the reviewer for their assessment of our work; we are pleased that they agree that the relevance of this work is extensive, and that general readership of Nature Communications will be interested.

1. The manuscript contains some typos: E.g. Page 8 'JDOS' Is it really that - if so, it is only vaguely defined as joint (J) density of states (DOS) once, or 'DOS?', page 8 'equilibrated' - please check the manuscript carefully.

We have gone through the manuscript carefully, and remedied some typos - we thank the reviewer for pointing this out. Regarding the two examples given: (i) we have clarified the use of 'equilibrated' (page 8), replacing it with 'not reached energetic equilibrium', and (ii) our delineation of JDOS (as pertaining to excitons) and DOS (as pertaining to polarons) in disordered systems is an important one; while the two are intimately related, our steady-state optical spectroscopy (such as PDS) is a probe of the former. Of course, measuring the PDS spectrum (as we have done) is not intended as a holistic mapping of the JDOS, but in disordered materials of comparable structure, we believe a measurement of the absorption cross section of tail states provides a reasonable approximation of the disorder in the JDOS. Our methodology aligns with various other recent publications (Kronemeijer et al. *Adv. Mater.*, **26**, 5, 728; Venkateshvaran et al, *Nature*, **515**, 7527, 384; Paleari et al. *Appl. Phys. Lett.*, **91**, 14, 141913)

2. Determination of PLQE: The authors state that the PLQE is determined for encapsulated samples. How is the encapsulation made? Typically, the Encapsulation induces significant absorption by the glue used that hampers the PLQE values.

We find that the PL spectrum of the epoxy resin used for encapsulation has vibronic peaks at 450nm, 486nm and 530nm under CW illumination at 405nm. Due to the spectral separation of this with that of our polymer PL (which begins at 565nm for TIF-F2BT), and prior measurement of the PLQE of the resin (in a neat resin film), we are able to deduce the absorption due to the resin itself in situ, and subtract it in the calculation.

3. Figure 4: The figure caption is missing information for panels d, e, and f. Furthermore, why is the distance for the panels c and d given, but not for e and f. And, very important, the figure caption does not include the description what the panels c-f are showing. Please add the information that there are each two polymer chains crossing. Otherwise, the reader has a hard time to figure this out. // Also, how are the distributions determined. Add information to the caption.

We thank the reviewer for drawing our attention to this: it was an oversight by us, and we have added text the figure caption to explain that c-f show 'various model geometries for the close crossing of two polymer chains'. Further to this, we have added the missing information for panels d-f, and clarified the distances for e-f by printing them on the figure for clarity.

Regarding how the distributions are determined, this is rather too complicated for the figure caption, and deserving of its own section in the supplementary information. During our discussion of Figure 4a in the main text, we have now added a clearer reference to Section S4 for the interested reader.

4. Figure 1: There are two problems here: a) The materials part of this study (colored dots) are not labelled, so there is no way to find out, which data point corresponds to which actual material. b) The literature values are all labeled with a grey triangle. While this may be done to give the overall trend, it is important for the reader to know the origin of each data set. Either include this into this figure, or add an SI plot, disclosing all literature sources.

We agree strongly with the second point made (that the origin of each datapoint should be clear), and have added a Table S1 citing the sources of the values plotted in the main text. Furthermore, we have clarified where these values are measured by us, and this is reflected both in Table S1 and in Figure 1 of main text. As before, we highlight the family of polymers which are the focus of this paper by colouring them in the same colours as in subsequent figures.

Regarding the first point (that the coloured dots are not labelled); this is intentional. The reviewer infers correctly that we wish to emphasise how our polymers fit into the general trends, and not to draw attention to any particular material at this stage of the paper; indeed, we have not even introduced the names of our polymers at this stage. However, we appreciate that the interested reader may wish to identify individual materials on the plot, and this is made possible by (a) the addition of Table S1, and (b) that materials are printed with a common colour-code throughout the entire manuscript in every figure and also in the SI.

5. On page 9, the authors state: 'Separately, we have observe similar dynamics for IDT-H2BT in unpublished work.' This is unsatisfying for the reader. Can't the essence of this work be included in the SI? A high level publication such as in Nature Communications should allow this extra detail.

We appreciate this suggestion and have included a reference to the PhD thesis of Tudor Thomas. This contains extensive characterization on other materials including IDT-H2BT, which goes beyond the scope of the present paper.

6. Figure 5: Again, no label to the actual molecules. Please include proper legends.

This has been amended.

7. Figure 5: For the ϕ_{PL} , it would be better to have the y-axis scaled all the way to 1 (unity), as this represents the upper limit. And further, put both k_r and k_{nr} axis to the same ranges, to better compare.

We thank the reviewer for this feedback, and agree that the y-axis on the top panel should be scaled to unity to demonstrate the theoretical upper limit; this has been amended in the figure. Furthermore, we have merged

the lower two panels to show k_r and k_{nr} on the same ranges for easy comparison, also amending the main text according with this change.

Reviewer #3 (Remarks to the Author):

In this manuscript, the authors reported a conjugated polymer of both high hole mobility and fluorescence quantum efficiency, which may foresee applications in electrically excited polymer lasers. Comparative study of a series of conjugated polymers, including model polymer IDT-H2BT, backbone fluorinated and pi-elongated derivatives, has demonstrated that interchain “close-crossing points” not only facilitate percolation of charge carries in the polymer network, but also provide emissive species of long lifetime. Finally, the highest $\Phi \cdot \mu$ value was obtained from TIF-H2BT via proper interplay between mobility and luminescence. This is basically an interesting and enlightening research considering the theoretical and experimental characterizations, I would recommend publication of this manuscript but after minor revision by the authors by addressing the following points:

We thank the reviewer for the positive remarks.

1. The contact resistance was mentioned across the electrical characterization section, however, experimental data of contact resistance were not included and should be given. As inferred by the output characteristics, the difference between the contact resistance of TIF-H2BT and IDT-H2BT was surprisingly small as for ionization potential 5.7 eV of TIF-H2BT (Adv. Mater. 2017, 29, 1702523) vs 5.4 eV of IDT-H2BT (J. Am. Chem. Soc., 2010, 132 (33), pp 11437–11439), and surprisingly large as for IP 5.7 eV of TIF-H2BT vs 5.8 eV of TIF-F2BT (Adv. Mater. 2017, 29, 1702523), given Au contacts were not modified to improve injection.

Our fabrication technique is able to improve charge injection significantly by incorporating molecular additives, which passivate water induced traps as discussed in Section S2. The uniformity and contact resistance of a range of polymers can be optimized in this way (Ref. S12) and this is in our experience more effective than contact modification with self-assembled monolayers.

This being said, the very deep lying HOMO levels (5.7 - 5.8 eV) of TIF-H2BT, TIF-FBT and TIF-F2BT will inevitably result in a larger injection barrier as compared to IDT-H2BT (HOMO of 5.3 eV). Furthermore, their deeper HOMO levels make the TIF polymers more sensitive to trapping of polarons on the polymer backbone by residual water molecules. This makes it even more important to incorporate molecular additives into the films than in IDT-H2BT.

We attempted to extract the contact resistance using the Transfer Length Method (TLM) for TIF-FBT and although we can extract a contact resistance value that is only slightly higher than in IDT-H2BT (fabricated using our additive route as well), the device uniformity is not good enough to extract precise values. The same applies to the polymers TIF-BT and TIF-DFBT. We show an example of the TLM analysis for TIF-FBT and IDT-H2BT in new Figure S1.

2. As shown in Figure S1, molecular weights and PDIs varied within this series of polymers. It should be discussed whether the electrical and photophysical properties are dependent on these two factors.

For a related polymer (IDT-BT with hexadecylbenzyl sidechains), we have investigated the molecular weight dependence in detail and found that a higher degree of polymerisation does not appreciably increase or decrease the mobility (new Ref. 66). Furthermore, we do not observe any trend in the mobility with PDI. This is consistent with earlier results in IDT-BT which showed that the mobility saturates once the molecular weight exceeds about 20 kDa. In the present work all polymers have reasonably high molecular weights above 45kDa we do not expect a significant dependence of the electrical and photophysical properties on molecular weight in this class of polymers.

REVIEWERS' COMMENTS:

Reviewer #1 (Remarks to the Author):

In manuscript NCOMMS-18-29533A, Thomas and co-authors demonstrate by extending the donor unit length in D-A copolymers can lead to conjugated copolymers with simultaneously high carrier mobility and luminescence quantum yield. This topic is appealing to the community of organic electronics due to its potential value for high-performing light-emitting transistors. The rule uncovered the authors that a trend of widened bandgaps and increased luminescence yield from IDT, IDTT to TIF-containing copolymers are of value for light-emitting material screening. Also, the photo-physical characterization of these materials are convincing. The authors have addressed my comments well and I recommend the publication of this research in Nature Communications.

Reviewer #2 (Remarks to the Author):

In the present version of the manuscript (NCOMMS-18-29533A), the authors present a major revision of their original manuscript. All points raised during the original review round have been addressed in a detailed and appropriate way, so that no points remain open. Thanks a lot, this new version has become much clearer for the reader. Hence, I would like to recommend this revised manuscript for acceptance.

Sebastian Reineke

Reviewer #3 (Remarks to the Author):

The revised manuscript and supplemental files read well in this version, and all my concerns have been addressed. I recommend acceptance at this time.